# RetroDiff: Retrosynthesis as Multi-stage Distribution Interpolation

## Abstract

Retrosynthesis poses a fundamental challenge in biopharmaceuticals, aiming to aid chemists in finding appropriate reactant molecules and synthetic pathways given determined product molecules. With the reactant and product represented as 2D graphs, retrosynthesis constitutes a conditional graph-to-graph generative task. Inspired by the recent advancements in discrete diffusion models for graph generation, we introduce Retrosynthesis Diffusion (RetroDiff), a novel diffusion-based method designed to address this problem. However, integrating a diffusion-based graph-to-graph framework while retaining essential chemical reaction template information presents a notable challenge. Our key innovation is to develop a multi-stage diffusion process. In this method, we decompose the retrosynthesis procedure to first sample external graph motifs from the dummy distribution given products and then generate the external bonds to connect the products and generated motifs. Interestingly, such a generation process is exactly the reverse of the widely adapted *semi-template* retrosynthesis procedure, *i.e.* from reaction center identification to synthon completion, which significantly reduces the error accumulation. Experimental results on the benchmark have demonstrated the superiority of our method over all other *semi-template* methods.

## 1 Introduction

Retrosynthesis (Corey, 1991) is an important topic in organic synthesis, which aims to help chemists find legitimate reactant molecules and synthetic pathways given product molecule, thus providing efficient and stable drug discovery and compound preparation methods for the biopharmaceutical field. Since the first computer-aided approaches were investigated (Corey & Wipke, 1969), huge efforts have been devoted in this area to explore analytical computational methods for retrosynthesis planning, and the research for data-driven methods has reached its peak in recent years with the machine learning boom.

Among the recent progress, the retrosynthesis methods can be broadly categorized into three groups. The *template-based* methods aim to retrieve the best match reaction template for a target molecule from a large-scale chemical database (Schneider et al., 2016; Somnath et al., 2020; Chen & Jung, 2021). Though with appealing performance, the scalability of the template-based method is indeed limited by the template database size (Segler & Waller, 2017; Segler et al., 2018). The *template-free* methods instead aim to generate the reactants given corresponding products directly without the involvement of the chemical prior (Zheng et al., 2019; Seo et al., 2021; Tu & Coley, 2022). While limited chemical reaction diversity and interpretability hinder the potential of template-free methods in practical applications (Chen et al., 2019; He et al., 2018; Jiang & de Rijke, 2018; Roberts et al., 2020).

Fortunately, the *semi-template* methods could be another alternative for building retrosynthesis models. Combining the strengths of both previously mentioned template-based and -free methods, the semi-template method introduces the chemical prior into the model design by employing a two-stage process including reaction center prediction and synthon completion. This makes the semi-template method more scalable than the template-based one and more interpretable than the template-free one, which has drawn increasing interest of late (Yan et al., 2020; Shi et al., 2020; Wang et al., 2021). In this paper, our goal is to develop a more efficient semi-template method for retrosynthesis.

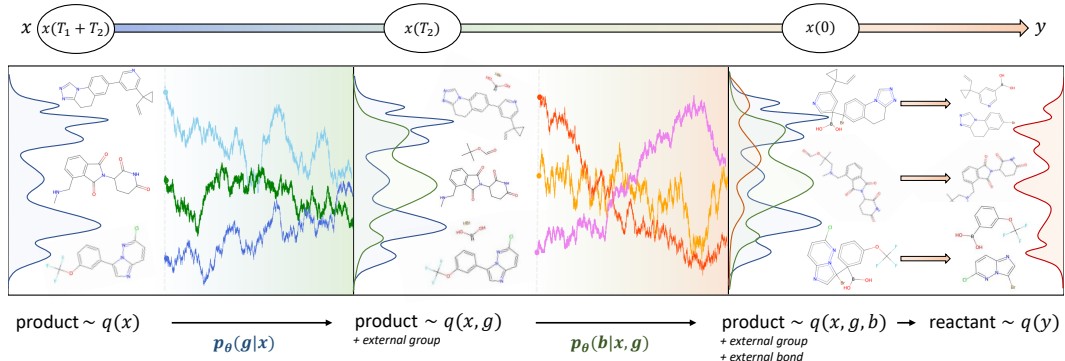

$x$ $x(T_1 + T_2)$ —— $x(T_2)$ —— $x(0)$ → $y$

product $\sim q(x)$ $\xrightarrow{\phantom{xxx}}$ product $\sim q(x,g)$ $\xrightarrow{\phantom{xxx}}$ product $\sim q(x,g,b)$ → reactant $\sim q(y)$

$\boldsymbol{p_\theta(g|x)}$ $\quad$ + external group $\quad\quad$ $\boldsymbol{p_\theta(b|x,g)}$ $\quad$ + external group
$\quad\quad\quad\quad\quad\quad\quad\quad\quad\quad\quad\quad\quad\quad\quad\quad\quad\quad\quad\quad\quad\quad\quad\quad$ + external bond

Figure 1: The overview and examples of our RetroDiff model.

The non-autoregressive diffusion generative model is particularly well-suited for capturing the complex structure of graph data, along with its robust capability for probabilistic modeling. Nonetheless, strictly following the conventional chemical reaction template can restrict the model's exploratory power with respect to distribution transformation. This is because it overly constrains the intrinsic data structure, necessitating artificial modifications to the molecular structure of groups and bonds.

To address this issue, we redefine the reaction template by separating the generation of external groups from the generation of bonds. This revised approach aligns with the concept of retrosynthesis, wherein the task is to transform distributions with minimal constraints: given a product molecule, we generate a "dummy" distribution that transitions to distributions of external groups and bonds. We then splice these to form the distribution of reactants. The traditional chemical reaction template is not suitable for this transformation, particularly in predicting the reaction center. Hence, our reformulation facilitates this process by distinguishing the generation of external groups from that of bonds, better accommodating the needs of distribution transformation within the modeling framework.

Building on this framework, we introduce **RetroDiff**—a **Retro**synthesis **Diff**usion model that works in discrete conditions. As illustrated in Figure 1, the model generates molecular structures through a two-stage denoising process. Initially, it begins with a simple distribution, proceeding to first create the external groups, which are parts that attach to the core molecule. Once these groups are formed, the model then constructs the bonds that connect these external groups to the product. In the final step, we adjust the molecule by removing certain bonds based on the atom's bonding capacity (valence), thereby ensuring the resulting reactant is chemically valid. Effectively, this is similar to a post-processing step where the reaction center is dissected. This proposed approach flips the script on the conventional semi-template method for retrosynthesis. Typically, the less uncertain task is performed first in order to minimize the buildup of errors.

Extensive experiment has been conducted on the benchmark dataset USPTO-50k (Schneider et al., 2016) under the semi-template setting. Our model achieves state-of-the-art performance compared with other competitive semi-template methods which further demonstrates the effectiveness of the proposed framework. Overall, our main contributions are three-fold:

- We propose RetroDiff, a multi-stage conditional retrosynthesis diffusion model that maps the product distribution to the reactant distribution. To this end, we decompose the reactant as a joint of the external group and external bonds and conduct the diffusion in the two variables in order.
- We redefine the pipeline of the semi-template methods by decomposing the task into external group generation and external bond generation to maximize the usage of chemical information in the molecule and reduce the error accumulation by determining the high entropy variable first.
- Our approach achieves state-of-the-art performance on the USPTO-50K dataset under the semi-template setting, and the reaction center prediction accuracy is significantly improved due to the acquisition of chemical information about the external groups.

## 2 RetroDiff: Retrosynthesis Diffusion

We begin by defining the task of retrosynthesis prediction. Consider a chemical reaction expressed as $\{\boldsymbol{G}_R^i\}_{i=1}^{|R|} \rightarrow \{\boldsymbol{G}_P^i\}_{i=1}^{|P|}$, where $\boldsymbol{G}_R$ represents the set of reactant molecular graphs, $\boldsymbol{G}_P$ represents the set of product molecular graphs, and $|R|$ and $|P|$ indicate the respective counts of reactants and products in a given reaction. Typically, we assume $|P| = 1$, which aligns with the conventions of benchmark datasets. The key problem in the retrosynthesis task is to invert the chemical reaction; namely deduce the reactant set $\{\boldsymbol{G}_R^i\}_{i=1}^{|R|}$ when presented with a sole product $\{\boldsymbol{G}_P\}$. In general, the assorted connected sub-graphs comprising the reactants can be amalgamated into a single disjoint graph $\{\boldsymbol{G}_R\}$. Thus, the retrosynthesis prediction problem simplifies to the transformation $\{\boldsymbol{G}_P\} \rightarrow \{\boldsymbol{G}_R\}$.

Existing semi-template retrosynthesis approaches typically first identify the reaction center in the target product and then complete the corresponding synthons at the fractured site. However, such a template setup is infeasible for designing the appropriate generative diffusion process. To address this, we redefine the task template with the following preliminary notations: $\mathbf{x} \sim P_{\mathcal{X}}$ denotes the variable of product graphs and the corresponding distribution, $\mathbf{y} \sim P_{\mathcal{Y}}$ for the reactant variable, $\mathbf{g} \sim P_{\mathcal{G}}$ as the external group, and $\mathbf{b} \sim P_{\mathcal{B}}$ denotes the external bond. We elaborate on the revised template task in the subsequent stages:

- **Stage 1: External Group Generation.** The process commences with the generation of the external group $\mathbf{g}$ that will attach to the product $\mathbf{x}$. namely sampling from such distribution $P_{\mathcal{G}}(\mathbf{g}|\mathbf{x}; \theta)$ which parameterized by the neural network $\theta$.
- **Stage 2: External Bond Generation.** Next, the process involves the generation of the external bond $\mathbf{b}$, which will link the product $\mathbf{x}$ with the newly formed external group $\mathbf{g}$. Here, we focus on modeling the distribution $P_{\mathcal{B}}(\mathbf{b}|\mathbf{g}, \mathbf{x}; \theta)$.
- **Stage 3: Post-Adaptation (Rule-Based).** The concluding phase involves a manual adjustment, breaking the reaction center in the product in line with valence rules to yield the final reactant $\mathbf{y}$. This transformation is depicted as $P_{\mathcal{Y}}(\mathbf{y}|\mathbf{b}, \mathbf{g}, \mathbf{x})$ which is a predetermined rule-based mapping.

Building on this framework, we introduce RetroDiff, a novel approach that models the aforementioned stages collectively within a unified diffusion model framework. The above stage-by-stage procedure essentially implies an autoregressive decomposition of the probabilistic model for approximating the conditional distribution:

$$P_{\text{model}}(\mathbf{y}|\mathbf{x}; \theta) = \int P_{\mathcal{G}}(\mathbf{g}|\mathbf{x}; \theta) P_{\mathcal{B}}(\mathbf{b}|\mathbf{g}, \mathbf{x}; \theta) P_{\mathcal{Y}}(\mathbf{y}|\mathbf{b}, \mathbf{g}, \mathbf{x}) \, d\mathbf{b} \, d\mathbf{g}, \tag{1}$$

which essentially represents the transformation between the product distribution and the reactant distribution.

### 2.1 RetroDiff Pipeline

In this section, we introduce the whole pipeline of the proposed RetroDiff which includes the detailed implementations and training procedures of $P_{\mathcal{G}}(\mathbf{g}|\mathbf{x}; \theta)$, $P_{\mathcal{B}}(\mathbf{b}|\mathbf{g}, \mathbf{x}; \theta)$ and $P_{\mathcal{Y}}(\mathbf{y}|\mathbf{b}, \mathbf{g}, \mathbf{x})$.

To start with, the diffusion process is utilized for modeling all the corresponding conditional distributions. For completeness, we elaborate on the details for parameterizing the conditional distribution with a diffusion process. We take $P_{\mathcal{G}}(\mathbf{g}|\mathbf{x}; \theta)$ as an example. Under the context of diffusion models, the dimensions of the input variable and output variable should be aligned. Hence we append a dummy variable $\mathbf{v}_1$ which makes the input as $(\mathbf{v}_1, \mathbf{x})$; Correspondingly, the output is as $(\mathbf{g}, \mathbf{x})$. Note that here we have $\dim(\mathbf{v}_1) = \dim(\mathbf{g})$. Similarly, for $P_{\mathcal{Y}}(\mathbf{y}|\mathbf{b}, \mathbf{g}, \mathbf{x})$, the input is $(\mathbf{v}_2, \mathbf{g}, \mathbf{x})$ while the output is as $(\mathbf{b}, \mathbf{g}, \mathbf{x})$. For the training objective, we only calculate the objective on the variables concerned, $\mathbf{g}$ in $P_{\mathcal{G}}(\mathbf{g}|\mathbf{x}; \theta)$ and $\mathbf{b}$ in $P_{\mathcal{B}}(\mathbf{b}|\mathbf{g}, \mathbf{x}; \theta)$. Strictly, our models implies a transformation on the joint space as:

$$\mathcal{X} \times \mathcal{V}_1 \times \mathcal{V}_2 \rightarrow \mathcal{X} \times \mathcal{G} \times \mathcal{V}_2 \rightarrow \mathcal{X} \times \mathcal{G} \times \mathcal{B} \rightarrow \mathcal{Y}, \tag{2}$$

Details of the generation pipeline could be found in Figure 2. To simplify the representation, we denote the condition at each stage as $c$.

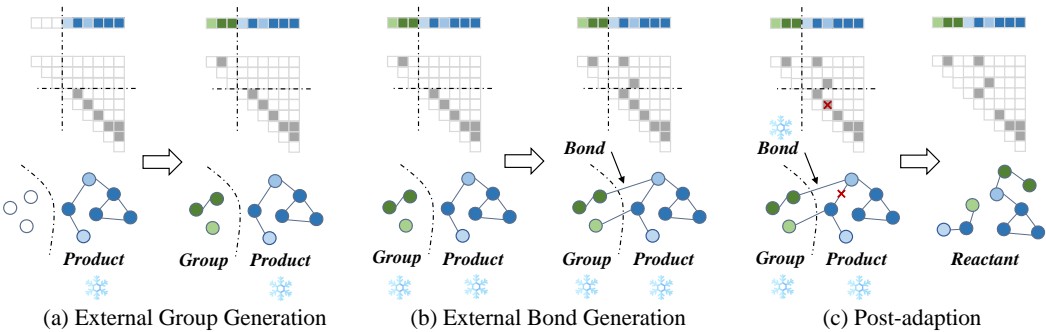

(a) External Group Generation    (b) External Bond Generation    (c) Post-adaption

Figure 2: The generation overview of the distribution transformation upon our template. The top row indicates changes in the atom types in the graph, the middle row indicates changes in the adjacency matrix of the graph, and the bottom row indicates overall changes in the graph structure. Specifically, the hollow circle denotes a dummy atom category we set for this task, and the colored circles denote real atom categories. The Line between circles means there exists one bond between the two atoms.

### 2.1.1 EXTERNAL GROUP GENERATION

The goal of this stage is to interpolate the distribution $P_{\mathcal{V}_1}$ to $P_{\mathcal{G}}$ conditional on $c$. In this stage, condition $\mathbf{c}$ is the product $\mathbf{x} \sim P_{\mathcal{X}}$. This is a **graph-to-graph** generative process, we define $\mathbf{v} \sim P_{\mathcal{V}_1}$ as a dummy graph and $\mathbf{g} \sim P_{\mathcal{G}}$ as an external group.

**Noise-applying.** In the noise-applying process, we interpolate the distribution $P_{\mathcal{G}}$ to $P_{\mathcal{V}_1}$ conditional on $\mathbf{c}$. With a slight abuse of notation, we splice external group $\mathbf{g}$ and product $\mathbf{x}$ as one unconnected graph $\boldsymbol{G} = (\boldsymbol{X}, \boldsymbol{E})$ with $n$ atoms and $m$ bonds, each atom and bond have $a$ and $b$ categories, respectively, so they can be represented by one-hot attributes that $\boldsymbol{X} \in \mathbb{R}^{n \times a}$ and $\boldsymbol{E} \in \mathbb{R}^{n \times n \times b}$. For graph $\boldsymbol{G}$, each atom and bond are diffused independently, which means the state transition each time acts on the single atom $x_i \in \boldsymbol{X}$ and bond $e_i \in \boldsymbol{E}$.

We follow Austin et al. (2021) to define the probability transitional matrix $\boldsymbol{Q}_t$ to conduct state transitions at each time $t$ in the discrete space. For graph $\boldsymbol{G}$, we apply noise to them via Markov matrices $[\boldsymbol{Q}_t^X]_{ss'} = q(x_t = s'|x_{t-1} = s)$ (for atom) and $[\boldsymbol{Q}_t^E]_{ss'} = q(e_t = s'|e_{t-1} = s)$ (for bond), where $s$ and $s'$ represent the state of atom (or bond) at time $t-1$ and $t$, respectively. Due to the graph independence, the noise-applying process for graph $\boldsymbol{G}$ can be defined as:

$$q(\boldsymbol{G}_t|\boldsymbol{G}_{t-1}) = (\boldsymbol{X}_{t-1}\boldsymbol{Q}_t^X, \boldsymbol{E}_{t-1}\boldsymbol{Q}_t^E) \implies q(\boldsymbol{G}_t|\boldsymbol{G}_0) = (\boldsymbol{X}_0\bar{\boldsymbol{Q}}_t^X, \boldsymbol{E}_0\bar{\boldsymbol{Q}}_t^E), \tag{3}$$

where $\boldsymbol{G}_0$ is the graph of ground truth, $\bar{\boldsymbol{Q}}_t^X = \prod_{i=1}^{t} \boldsymbol{Q}_t^X$ and $\bar{\boldsymbol{Q}}_t^E = \prod_{i=1}^{t} \boldsymbol{Q}_t^E$. Finally, we sample the probability distribution $q(\boldsymbol{G}_t|\boldsymbol{G}_0)$ to obtain the noisy graph $\boldsymbol{G}_t$.

**Denoising.** In the denoising process, given a noisy graph $\boldsymbol{G}_t$ and condition $c$, we need to iterate the denoising process $p_\theta(\boldsymbol{G}_{t-1}|\boldsymbol{G}_t, c)$ by a trainable network $p_\theta$ at each time $t$. We model the distribution as the product over nodes and edges and marginalize each item over the network predictions:

$$p_\theta(\boldsymbol{G}_{t-1}|\boldsymbol{G}_t, \mathbf{c}) = \prod_{x \in \boldsymbol{X}_{t-1}} p_\theta(x|\boldsymbol{G}_t, \mathbf{c}) \prod_{e \in \boldsymbol{E}_{t-1}} p_\theta(e|\boldsymbol{G}_t, \mathbf{c}),$$

$$\text{where} \quad p_\theta(x|\boldsymbol{G}_t, \mathbf{c}) = \sum_{x_0 \in \boldsymbol{X}_0} q(x|x_t, x_0, \mathbf{c})p_\theta(x_0|\boldsymbol{G}_t, c), \tag{4}$$

$$p_\theta(e|\boldsymbol{G}_t, \mathbf{c}) = \sum_{e_0 \in \boldsymbol{E}_0} q(e|e_t, e_0, \mathbf{c})p_\theta(e_0|\boldsymbol{G}_t, c).$$

Next, we derive $q(\boldsymbol{G}_{t-1}|\boldsymbol{G}_t, \boldsymbol{G}_0, \mathbf{c})$ with the Bayes theorem and transform it into forms of node and edge to complete the calculations in Eq.(4) (Vignac et al., 2022). For node $X$, we have:

$$q(\boldsymbol{X}_{t-1}|\boldsymbol{X}_t, \boldsymbol{X}_0, \mathbf{c}) = \frac{q(\boldsymbol{X}_t|\boldsymbol{X}_{t-1}, \boldsymbol{X}_0, \mathbf{c})q(\boldsymbol{X}_{t-1}|\boldsymbol{X}_0, c)}{q(\boldsymbol{X}_t|\boldsymbol{X}_0, \mathbf{c})}$$

$$= \frac{\boldsymbol{X}_t[\boldsymbol{Q}_t^X]^\top \odot \boldsymbol{X}_0\bar{\boldsymbol{Q}}_{t-1}^X}{\boldsymbol{X}_0\bar{\boldsymbol{Q}}_t^X[\boldsymbol{X}_t]^\top} \propto \boldsymbol{X}_t[\boldsymbol{Q}_t^X]^\top \odot \boldsymbol{X}_0\bar{\boldsymbol{Q}}_{t-1}^X, \tag{5}$$

Similarly, $q(\boldsymbol{E}_{t-1}|\boldsymbol{E}_t, \boldsymbol{X}_0, \mathbf{c}) \propto \boldsymbol{E}_t[\boldsymbol{Q}_t^E]^\top \odot \boldsymbol{E}_0\bar{\boldsymbol{Q}}_{t-1}^E$. Based on the above derivation, we only need to create a network $\hat{p}(\boldsymbol{G}_0|\boldsymbol{G}_t, \mathbf{c})$ to predict clean graph $\boldsymbol{G}_0$ given noisy data $\boldsymbol{G}_t$ and condition $\mathbf{c}$.

### 2.1.2 EXTERNAL BOND GENERATION

In this stage, we aim to interpolate the distribution $P_{\mathcal{V}_2}$ to $P_{\mathcal{B}}$ conditional on $\mathbf{c}$, where condition $\mathbf{c}$ is $P_{\mathcal{X}} \times P_{\mathcal{G}}$. This is a **bond-to-bond** generative process, we define $\mathbf{v} \sim P_{\mathcal{V}_2}$ as a dummy bond set and $\mathbf{b} \sim P_{\mathcal{B}}$ connecting $\mathbf{g}$ and $\mathbf{x}$. In the same way, we splice $\mathbf{g}$, $\mathbf{x}$, and $\mathbf{b}$ as a connected graph. We have obtained a trained network $p_\theta$ in the last stage, so we freeze $\mathbf{g}$ and $\mathbf{x}$ in the graph and continue to train $p_\theta$.

### 2.1.3 POST-ADAPTION

Under traditional semi-template methods, the reaction center in the product is first predicted. After breaking the bond, completion is conducted at the reaction site. In the first two stages, we equivocally complete the "completion", and thus need to break the reaction bond to obtain a legitimate reactant finally. There exist two possible situations: without and with broken bonds. If without broken bonds, only one external bond connects the product to the external group. If with broken bonds, two external bonds connect two atoms in the product and these two atoms are connected to each other by a legal bond.

We reverse this observation to design a post-adaption rule. Specifically, when only one external bond is generated, no treatment of the product is required. When two external bonds are generated, if there is a legal bond between the two product atoms connected by the two external bonds, they are broken, otherwise, they are judged to be illegally generated. Figure 3 shows the valid and invalid situations.

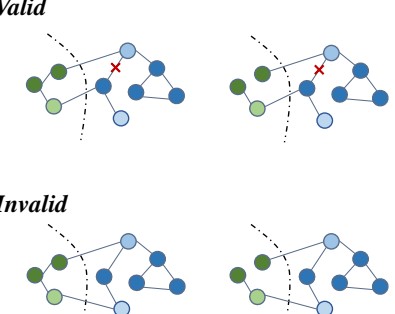

**Valid**

**Invalid**

Figure 3: Valid and invalid situations of the post-adaption operation.

### 2.2 PRIOR DISTRIBUTION AND INTERPOLATION DIRECTION

In this part, we design the task-specific prior distribution (i.e. sampling start) $P_{\mathcal{V}_1}$ and $P_{\mathcal{V}_2}$ in the first two stages, and interpolation direction (i.e. transitional matrix) $\boldsymbol{Q}_t^X$ and $\boldsymbol{Q}_t^E$. We denote $n_g$ and $n_x$ as the atom numbers of the external group $\mathbf{g}$ and the product $\mathbf{x}$, respectively. It is noted that we cannot predict the exact atom number of external groups in different cases, so we restrict $n_g$ as a constant and create a dummy atom category [1]. We set $n_g$ as the maximum real atom number of all external groups in the dataset.

**Prior Distribution.** All atoms can start from a single distribution $v_x$ and all bonds can start from $v_e$. In the external group generation of stage 1, both atoms and bonds need to be denoised, but in the external bond generation of stage 2, only bonds need to be denoised. Therefore, the two prior distributions can be formulated as

$$P_{\mathcal{V}_1} = \prod_{1 \le i \le |n_g|} p_{v_x} \times \prod_{\substack{1 \le i \le |n_g| \\ 1 \le j \le |n_g|}} p_{v_e}, \qquad P_{\mathcal{V}_2} = \prod_{\substack{1 \le i \le |n_g| \\ 1 \le j \le |n_x|}} p_{v_e}. \tag{6}$$

For all atoms and bonds samples from the dummy state, we set the probability of single distribution as $p_{v_x} = [1, 0, 0, ..., 0]^\top \in \mathbb{R}^{1 \times (a+1)}$ and $p_{v_e} = [1, 0, 0, ..., 0]^\top \in \mathbb{R}^{1 \times (b+1)}$, where the first position

---

[1] As an example, at the sampling stage, the external group starts from $n_g$ dummy atoms, and half of them are denoised to the real atoms, then in this case, the actual number of external group atoms is $n_g/2$.

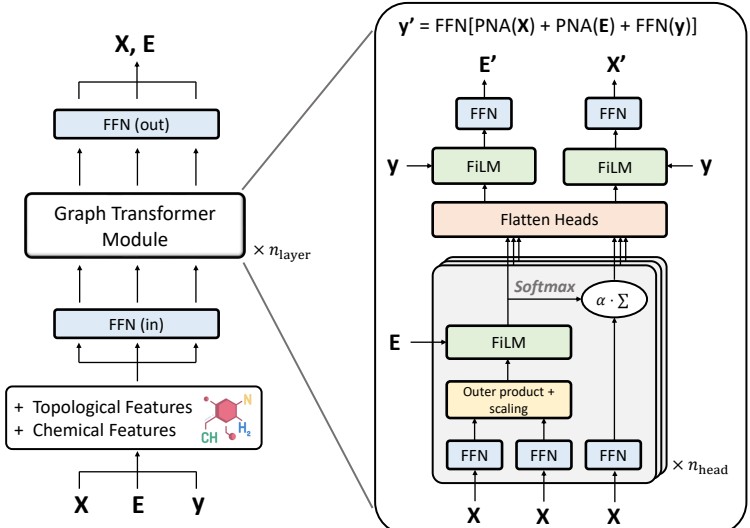

Figure 4: The whole architecture (left) of the denoising network for training with graph transformer modules (right). $\boldsymbol{X}, \boldsymbol{E}, \boldsymbol{y}$ denote the atom features, bond features, and global features, respectively. $\text{FiLM}(\boldsymbol{M_1}, \boldsymbol{M_2}) = \boldsymbol{M_1}\boldsymbol{W_1} + (\boldsymbol{M_2}\boldsymbol{W_2}) \odot \boldsymbol{M_2} + \boldsymbol{M_2}$, where $\boldsymbol{W_1}, \boldsymbol{W_2}$ are learnable. $\text{PNA}(\boldsymbol{M}) = [\max(\boldsymbol{M}) \circ \min(\boldsymbol{M}) \circ \text{mean}(\boldsymbol{M}) \circ \text{std}(\boldsymbol{M})]\boldsymbol{W}$, where $\boldsymbol{W}$ is learnable.

in the vector denotes the dummy atom (bond) category and the other positions denote each real categories ($a$ types of atoms and $b$ types of bonds), respectively.

**Interpolation Direction.** For the diffusion model to be reversible, any sample $s = (s_x, s_e) \sim p_{\text{data}}$ ($p_{\text{data}}$ denotes the whole data distribution) must converge to a limit distribution $q_\infty$ after $t$-step noise-applying, i,e., $q_\infty = \lim_{t\to\infty} s\bar{\boldsymbol{Q}}_t$, which in turn is the sampling start. Therefore, we need to design $\boldsymbol{Q}_t^X$ and $\boldsymbol{Q}_t^E$ to satisfy that for any atom $s_x$ and bond $s_e$ from the data distribution, $v_x = \lim_{t\to\infty} s_x\bar{\boldsymbol{Q}}_t^X$ and $v_e = \lim_{t\to\infty} s_e\bar{\boldsymbol{Q}}_t^E$. Considering $s_x$ and $s_e$ are one-hot vectors, we compute $\lim_{t\to\infty} \bar{\boldsymbol{Q}}_t^X = \mathbf{1}_x v_x^\top$ and $\lim_{t\to\infty} \bar{\boldsymbol{Q}}_t^E = \mathbf{1}_e v_e^\top$, so a trivial design is

$$\boldsymbol{Q}_t^X = \alpha_t \boldsymbol{I} + (1 - \alpha_t)\mathbf{1}_x v_x^\top, \qquad \boldsymbol{Q}_t^E = \alpha_t \boldsymbol{I} + (1 - \alpha_t)\mathbf{1}_e v_e^\top, \tag{7}$$

where $\boldsymbol{I}$ is an identity matrix, $\mathbf{1}_x$ and $\mathbf{1}_e$ are all-one vectors, $\alpha_t$ is a cosine schedule that $\bar{\alpha}_t = \cos{(0.5\pi(t/T + s)/(1 + s))}^2$ (Nichol & Dhariwal, 2021).

## 2.3 DENOISING NETWORK FOR TRAINING

We design $p_\theta(\boldsymbol{G}_0|\boldsymbol{G}_t, \mathbf{c})$ to model $p_\theta(\boldsymbol{G}_{t-1}|\boldsymbol{G}_t, \mathbf{c})$ at the above stages. Specifically, at 0 to $T_2$ steps, we take group $\mathbf{g}$, product $\mathbf{x}$, and noisy external bonds $b_t$ as the input, and clean external bond $b_0$ as the output. At $T_2 + 1$ to $T_2 + T_1$ steps, we take product $\mathbf{x}$, and noisy external group $\boldsymbol{G}_t$ as the input, and clean external group $\boldsymbol{G}_0$ as the output. The training loss of $p_\theta(\boldsymbol{G}_0|\boldsymbol{G}_t, \mathbf{c})$ can be formulated as:

$$\mathcal{L}(\boldsymbol{G}_0 = (\boldsymbol{X_0}, \boldsymbol{E_0})) = -\mu \cdot \sum_{x \in \boldsymbol{X}_0} \log(p_\theta(x)) - \sum_{e \in \boldsymbol{E}_0} \log(p_\theta(e)), \tag{8}$$

where $\mu$ is a control unit, in stage 1, $\mu = 0$, and in stage 2, $\mu$ is a hyperparameter to trade off the importance of atoms and bonds. In general, $\mu < 1$.

We use the graph transformer architecture (Vignac et al., 2022) to design the network. The overall architecture and the graph transformer module for each layer are shown in Figure 4. Specifically, we merge the input graph $\boldsymbol{G}_t$ at step $t$ and condition $\mathbf{c}$ into a whole graph structure $\boldsymbol{G} = (\boldsymbol{X}, \boldsymbol{E}, \boldsymbol{y})$, where $y$ is the global features, and obtain the topological features and chemical features (Details can be seen in Appendix A) of this molecular graph to splice with the original features. After the pre-processing, $\boldsymbol{G} = (\boldsymbol{X}, \boldsymbol{E}, \boldsymbol{y})$ is input to a feed-forward network to be encoded, then it will pass serially through the $n_{\text{layer}}$ graph transformer modules. Finally, another feed-forward network is set to decode the graph features, the output is the final prediction result.

Table 1: Top-$k$ accuracy for the retrosynthesis task on USPTO-50K dataset.

| Model | Top-$k$ accuracy | | | |
|---|---|---|---|---|
| | $k=1$ | $k=3$ | $k=5$ | $k=10$ |
| **Template-based methods** | | | | |
| GLN (Schneider et al., 2016) | 52.5 | 69.0 | 75.6 | 83.7 |
| GraphRetro (Somnath et al., 2020) | 53.7 | 68.3 | 72.2 | 75.5 |
| LocalRetro (Chen & Jung, 2021) | 53.4 | 77.5 | 85.9 | 92.4 |
| RetroKNN (Xie et al., 2023) | 55.3 | 76.9 | 84.3 | 90.8 |
| **Template-free methods** | | | | |
| Transformer (Vaswani et al., 2017) | 42.4 | 58.6 | 63.8 | 67.7 |
| SCROP (Zheng et al., 2019) | 43.7 | 60.0 | 65.2 | 68.7 |
| Transformer (*Aug.*) (Tetko et al., 2020) | 48.3 | - | 73.4 | 77.4 |
| Tied Transformer (Kim et al., 2021) | 47.1 | 67.1 | 73.1 | 76.3 |
| GTA (Seo et al., 2021) | 51.1 | 67.6 | 74.8 | 81.6 |
| Graph2SMILES (Tu & Coley, 2022) | 52.9 | 66.5 | 70.0 | 72.9 |
| Retroformer (Wan et al., 2022) | 47.9 | 62.9 | 66.6 | 70.7 |
| Retroformer (*Aug.*) (Wan et al., 2022) | 52.9 | 68.2 | 72.5 | 76.4 |
| RootAligned (Zhong et al., 2022) | 56.0 | 79.1 | 86.1 | 91.0 |
| **Semi-template methods** | | | | |
| RetroXpert (Yan et al., 2020) | 50.4 | 61.1 | 62.3 | 63.4 |
| G2G (Shi et al., 2020) | 48.9 | 67.6 | 72.5 | 75.5 |
| MEGAN (Sacha et al., 2021) | 48.1 | 70.7 | 78.4 | **86.1** |
| RetroPrime (Wang et al., 2021) | 51.4 | 70.8 | 74.0 | 76.1 |
| RetroDiff (ours) | **52.6** | **71.2** | **81.0** | 83.3 |

## 3 EXPERIMENTS

### 3.1 SETUP

**Dataset.** We perform experiments on the widely used USPTO-50K (Schneider et al., 2016), which contains 50,000 single-step chemical reactions from 10 reaction types. We follow standard splits to select 80% of data as the training set, 10% as the validation set, and 10% as the test set.

**Baseline.** Our baselines can be categorized into three groups: (i) *Template-based* methods, we choose GLN (Schneider et al., 2016), GraphRetro (Somnath et al., 2020), LocalRetro (Chen & Jung, 2021), and RetroKNN (Xie et al., 2023). (ii) *Template-free* methods, we choose Transformer (Vaswani et al., 2017), SCROP (Zheng et al., 2019), Tied Transformer (Kim et al., 2021), Augmented Transformer (Tetko et al., 2020), GTA (Seo et al., 2021), Graph2SMILES (Tu & Coley, 2022), Retroformer (Wan et al., 2022), and RootAligned (Zhong et al., 2022). (iii) *Semi-template* methods, we choose RetroXpert (Yan et al., 2020), G2G (Shi et al., 2020), RetroPrime (Wang et al., 2021), and MEGAN (Sacha et al., 2021).

**Implementation.** We use open-source RDKit to construct molecular graphs based on molecular SMILES. For noise-applying and sampling processes, we set $T_1 = 500$ and $T_2 = 50$. For the training process, we train the graph transformer at 8-card 24G GTX-3090 with a training step of 100000, a batch size of 120, and an Adam learning rate of 0.0001, and set $\mu = 0.2$. In addition, when setting $n_g$, during the statistical process, to avoid extreme values that cause sparse distributions, we exclude all samples whose statistic is more than three times the standard deviation from the mean.

**Evaluation.** We follow prior works to adopt top-$k$ accuracy as the main evaluation metric. For end-to-end models, beam search is adopted, but it is unfeasible for diffusion models. Therefore, we set the negative variational lower bound as the reranking score for each generated $G_0 = (X_0, E_0)$, yielding the following top-$k$ accuracy score:

$$\text{Score} = \mu \cdot \mathbb{E}_{q(x_0)}\mathbb{E}_{q(x_t|x_0)}[-\log p_\theta(x_0|x_t)] + \mathbb{E}_{q(e_0)}\mathbb{E}_{q(e_t|e_0)}[-\log p_\theta(e_0|e_t)]. \quad (9)$$

For each sampling, the smaller the score is, the closer the sample is to the true data distribution. We sample 100 results for each case to rerank the scores. In addition, we compute top-$k$ valid-

ity, which can reflect the legitimacy of the reactants as chemical molecules, and is formulated as $\frac{1}{N \times k} \sum_1^N \sum_1^k \text{isvalid}(\text{G}_0)$, where $N$ denotes the dataset size.

## 3.2 MAIN RESULTS

We report top-$k$ accuracy and validity in the reactant class unknown setting and compare our method with all strong template-based, template-free, and semi-template methods. Specifically, we categorize our method as a semi-template method.

Table 1 shows the top-$k$ accuracy results. Our method outperforms all other competitive *semi-template* baselines across different $k$ values, particularly excelling at top-5. In addition, our method demonstrates competitive results when compared to the strongest template-free methods, Graph2SMILES and Retroformer. Notably, our method holds a substantial advantage for $k > 1$. Additionally, we surpass the performance of GLN in the realm of template-based methods, underscoring the effectiveness of our template setup.

Table 2: Top-$k$ validity for the retrosynthesis task on USPTO-50K dataset comparing our method with some template-free methods.

| Model | Top-$k$ validity | | | |
|---|---|---|---|---|
| | $k = 1$ | $k = 3$ | $k = 5$ | $k = 10$ |
| Transformer | 97.2 | 97.9 | 82.4 | 73.1 |
| Graph2SMILES | **99.4** | 90.9 | 84.9 | 74.9 |
| Retroformer (*Aug.*) | 99.3 | 98.5 | 97.2 | 92.6 |
| RetroDiff (ours) | 99.2 | **99.0** | **97.8** | **94.3** |

Table 2 shows the top-$k$ validity results. We take the vanilla retrosynthesis Transformer, Graph2SMILES, and augmented Retroformer as strong template-free baselines (mainly on SMILES generation) for validity comparison with our model. We note that our validity score outperforms that of both Transformer and GraphSMILES, two end-to-end SMILES generation models. Despite Retroformer's competitive validity score as a template-free model, attributed to the integration of chemical information from the reaction center in their model, our method demonstrates a further improvement. This suggests that our model exhibits enhanced chemical feasibility,

## 3.3 ABLATION

In this part, We will conduct ablation studies to analyze the performances of generative sub-modules in each stage, namely external group generation and external bond generation.

### 3.3.1 EXTERNAL GROUP GENERATION

First, RetroDiff generates external groups given raw products. In traditional semi-template methods, the external group generation equates to a synthon completion task, commonly addressed through two distinct methods: (i) autoregressive generation, encompassing encoder-decoder sequence prediction (Shi et al., 2020) and action-state sequence prediction (Somnath et al., 2020), (ii) finite-space search, where all possible leaving group vocabularies are constructed using a database, followed by maximum likelihood estimation using

Table 3: Top-$k$ accuracy of the external group generation sub-module (* indicates the performance of raw synthon completion).

| Model | Top-$k$ accuracy | | | |
|---|---|---|---|---|
| | $k = 1$ | $k = 3$ | $k = 5$ | $k = 10$ |
| G2G* | 61.1 | 81.5 | 86.7 | 90.0 |
| RetroXpert* | 64.8 | 77.6 | 80.8 | 84.5 |
| RetroDiff (ours) | 66.5 | 78.4 | 85.0 | 86.4 |

a classification model (Yan et al., 2020). In our template setting, the external group generation is treated as a non-autoregressive generation task.

Table 3 shows the results and we compare the external group generation performance between our method and the synthon completion performance of other methods. Our external group generation outperforms the rest of the methods on top-1, but not as well as G2G when $k > 1$, albeit within a reasonable margin. A plausible explanation lies in the fact that G2G acquires information about the reaction center when generating the external group, i.e., serial complementation from the reaction sites. In contrast, our method may lack this specific information, resulting in a slight disadvantage.

### 3.3.2 EXTERNAL BOND GENERATION

Next, RetroDiff generates external bonds given products and generated external groups. In the traditional semi-template methods, reaction centers are predicted directly by the classification model, whereas under our template setup, this task equates to a combination of external bond generation and post-adaptation. Thus, we conduct a direct comparison between the performance of previous methods in predicting reaction centers and the external bond generation performance of our model. Table 4 shows the results, indicating that predicting the connecting bond between the product and the external group, and thus deducing the reaction center based on the rule, can lead to achieving higher accuracy than direct prediction of the reaction center given the product.

Table 4: Top-$k$ accuracy of the external bond generation sub-module (* indicates the performance of raw reaction center prediction).

| Model | Top-$k$ accuracy | | | |
|---|---|---|---|---|
| | $k=1$ | $k=3$ | $k=5$ | $k=10$ |
| G2G* | 61.1 | 81.5 | 86.7 | 90.0 |
| RetroXpert* | 64.3 | - | - | - |
| GraphRetro* | 75.6 | 87.4 | 92.5 | 96.1 |
| RetroDiff (ours) | 82.3 | 92.4 | 95.5 | 96.8 |

Specifically, the atom number of the product is denoted as $n$, the bond number as $m$, and the external group atom number as $c$ (with $c$ being a constant set to 10 in our experimental configuration). Considering a maximum bonding site limit of 4 for an atom (e.g., Carbon atom) excluding Hydrogen atoms, we establish the condition $m \leq 2n$. In the realm of traditional reaction center prediction, the search space size is $m$, whereas, for external bond generation, it is $cn$. Consequently, the complexity of the external bond generation task is higher than that of the reaction center prediction task. However, the external bond generation task leverages molecular information from external groups, expanding the model's ability to search for reaction sites more accurately by incorporating additional chemical insights. Consequently, the observed superior performance of external bond generation over traditional reaction center prediction can be empirically attributed to the enriched chemical information acquired through the former.

## 4 RELATED WORK

**Retrosynthesis Prediction.** Existing methods of retrosynthesis prediction can be broadly categorized into three groups: (i) *Template-based* methods retrieve the best match reaction template for a target molecule from a large-scale chemical database, they focus on computing the similarity scores between target molecules and templates using either plain rules (Coley et al., 2017) or neural networks (Schneider et al., 2016; Somnath et al., 2020; Chen & Jung, 2021). (ii) *Template-free* methods adopt end-to-end generative models to directly obtain final reactants given products (Zheng et al., 2019; Kim et al., 2021; Seo et al., 2021; Tu & Coley, 2022; Wan et al., 2022). Despite the efficiencies of data-driven methods, the chemical prior has been ignored. (iii) *Semi-template* methods combine the advantages of the above two approaches, they split the task into two parts, i.e., reaction center prediction and synthon correction (Yan et al., 2020; Shi et al., 2020; Wang et al., 2021), followed by serial modeling using a classification model and a generative model, respectively.

**Diffusion Models.** Diffusion models (Sohl-Dickstein et al., 2015; Ho et al., 2020) is a class of score-based generative models (Song & Ermon, 2019), whose goal is to learn the latent structure of a dataset by modeling the way in which data points diffuse through the latent space. Since the generalized discrete diffusion model (Austin et al., 2021) and the discrete graph diffusion model (Vignac et al., 2022) have been proposed, the molecular design field has begun to use them extensively, such as molecular conformation (Xu et al., 2021), molecular docking (Corso et al., 2022), and molecular linking (Igashov et al., 2022). To the best of our knowledge, we are the first to apply discrete diffusion models to the retrosynthesis prediction task.

## 5 CONCLUSION

We introduce RetroDiff, a multi-stage conditional retrosynthesis diffusion model. Considering maximizing the usage of chemical information in the molecule, we reset the template to decompose the retrosynthesis into external group generation and external bond generation sub-tasks, and set a joint diffusion model to transfer dummy distributions to group and bond distributions serially. Our method performs the best under the semi-template setting in the accuracy and validity evaluation metrics. In the future, we will try to extend our RetroDiff to multi-step retrosynthesis scenarios.

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

# A    ADDITIONAL FEATURES FOR NETWORK TRAINING

To fully explore the potential features of a molecular graph, we can analyze it from two perspectives: topological features and chemical features.

**Topological Features.**    We focus on two useful topological features. First is the **spectral features**, we first compute some graph-level features that relate to the eigenvalues of the graph Laplacian: the number of connected components (given by the multiplicity of eigenvalue 0), as well as the 5 first nonzero eigenvalues. We then add node-level features relative to the graph eigenvectors: an estimation of the biggest connected component (using the eigenvectors associated with eigenvalue 0), as well as the two first eigenvectors associated with non-zero eigenvalues.

Second is the **cycle detection**. To further refine it, we split it into node-level and graph-level features. For node-level features, we compute how many $k$-cycles this node belongs to, where $3 \leq k \leq 5$. The feature formulas are as follows:

$$
\begin{aligned}
\boldsymbol{X}_3 &= \mathrm{diag}(\boldsymbol{A}^3)/2, \\
\boldsymbol{X}_4 &= (\mathrm{diag}(\boldsymbol{A}^4) - \boldsymbol{d}(\boldsymbol{d}-1) - \boldsymbol{A}(\boldsymbol{d}\mathbf{1}_n^\top)\mathbf{1}_n)/2, \\
\boldsymbol{X}_5 &= (\mathrm{diag}(\boldsymbol{A}^5) - 2\mathrm{diag}(\boldsymbol{A}^3) \odot \boldsymbol{d} - \boldsymbol{A}(\mathrm{diag}(\boldsymbol{A}^3)\mathbf{1}_n^\top)\mathbf{1}_n + \mathrm{diag}(\boldsymbol{A}^3))/2,
\end{aligned}
\tag{10}
$$

where $\boldsymbol{d}$ denotes the vector containing node degrees. For graph-level features, we compute how many $k$-cycles this graph contains, where $3 \leq k \leq 6$. The feature formulas are as follows:

$$
\begin{aligned}
\boldsymbol{y}_3 &= \boldsymbol{X}_3^\top \mathbf{1}_n/3, \\
\boldsymbol{y}_4 &= \boldsymbol{X}_4^\top \mathbf{1}_n/4, \\
\boldsymbol{y}_5 &= \boldsymbol{X}_5^\top \mathbf{1}_n/5, \\
\boldsymbol{y}_6 &= \mathrm{Tr}(\boldsymbol{A}^6) - 3\mathrm{Tr}(\boldsymbol{A}^3 \odot \boldsymbol{A}^3) + 9\|\boldsymbol{A}(\boldsymbol{A}^2 \odot \boldsymbol{A}^2)\|_F - 6\mathrm{diag}(\boldsymbol{A}^2)^\top \mathrm{diag}(\boldsymbol{A}^4) \\
&\quad + 6\mathrm{Tr}(\boldsymbol{A}^4) - 4\mathrm{Tr}(\boldsymbol{A}^3) + 4\mathrm{Tr}(\boldsymbol{A}^2 \dot{\boldsymbol{A}}^2 \odot \boldsymbol{A}^2) + 3\|\boldsymbol{A}^3\|_F - 12(\boldsymbol{A}^2 \odot \boldsymbol{A}^2) + 4\mathrm{Tr}(\boldsymbol{A}^2),
\end{aligned}
\tag{11}
$$

where $\|\cdot\|_F$ is Frobenius norm.

**Chemical Features.**    There are two useful chemical features. First is the **atom valency**, which can be concatenated to the atom features $\boldsymbol{X}$. Second is the **molecular weight**, which can be concatenated to the global features $\boldsymbol{y}$.

# B    CASE STUDY VIA VISUALIZATION

In this section, we present visualizations of both successful and failed cases to provide an intuitive analysis of RetroDiff's mechanisms. Figure 5 illustrates instances of success, featuring external groups delineated by blue shaded boxes and external bonds highlighted in green. Conversely, Figure 6 showcases failed cases, revealing two prevalent situations associated with higher error rates: (i) elevated error rates are observed when the external group size is substantial, leading to biases in the prediction of bonds between atoms, and (ii) for external bond predictions, inaccuracies in predicting reaction sites on the product contribute to ineffective post-adaptation of reaction centers.

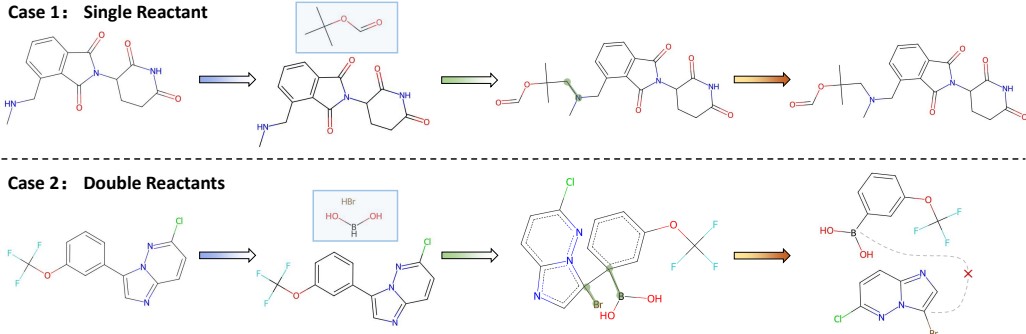

Figure 5: Successful cases produced by RetroDiff on the retrosynthesis task.

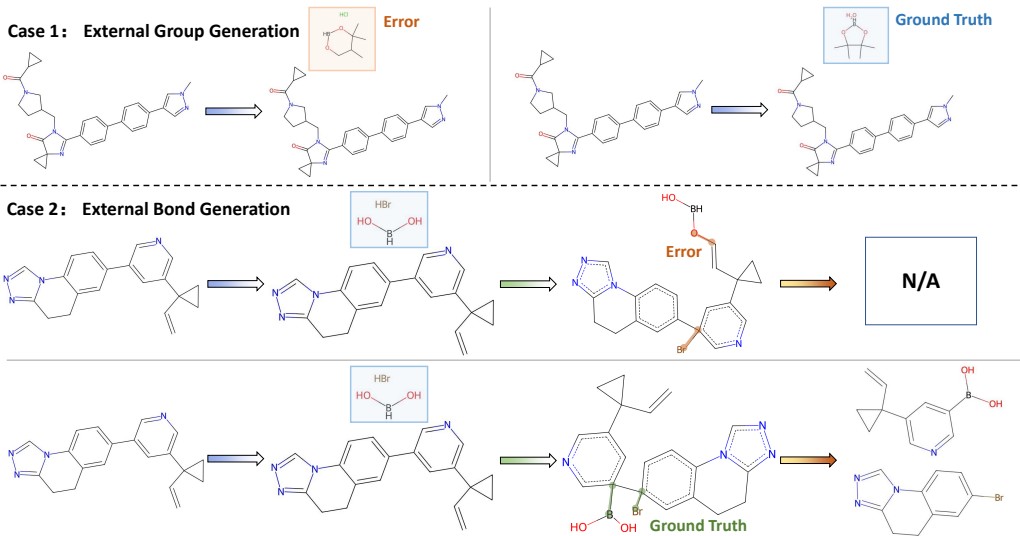

Figure 6: Failed cases produced by RetroDiff on the retrosynthesis task.

