# OpenReview forum: "RetroDiff: Retrosynthesis as Multi-stage Distribution Interpolation"
_ICLR.cc/2024/Conference — Submitted to ICLR 2024_

### Official Review · Reviewer_dttv · 2023-10-30

**Soundness:** 4 excellent
**Presentation:** 3 good
**Contribution:** 2 fair
**Rating:** 6
**Confidence:** 4

**Summary:**

This paper proposes a retrosynthesis prediction method under the semi-template setting which leverages some chemical logics. Specifically, whereas the existing semi-template methods consist of predicting the reaction center and then predicting the synthon completion, the authors decompose the retrosynthesis into three steps: 1) the external group generation, 2) the external bond generation, and 3) the post-adaption. The first two steps (i.e. the external group generation and the external bond generation) are modeled by the discrete denoising diffusion model, and in the last step (i.e. the post-adaption) the retrosynthesis is completed by leveraging the prior knowledge of the chemical reactions. The authors compare the proposed method with the template-based, template-free, and semi-template methods.

**Strengths:**

1. To my knowledge, this work is the first work to apply the diffusion model for planning the retrosynthesis.
2. By splitting the prediction of retrosynthesis into three stages, the proposed method is advantageous to reduce the search space. First, the external group generation and external bond generation have similar properties with the reaction center prediction and the synthon completion, which are an easier way to directly predict the retrosynthesis. Also, leveraging the chemical knowledge in the post-adaption stage is an efficient way to reduce the search space of the previous two stages.
3. The ablation study is insightful to understand the proposed method and well-designed to compare with the baselines.

**Weaknesses:**

1. The diffusion models usually require more time. In this paper, the two stage diffusion models could be less efficient than other baselines in terms of the inference time.
2. For the top-1 accuracy, the performance gain of the proposed method is marginal. However, it outperforms in top-3, 5, and 10 accuracy.

**Questions:**

1. How to preprocess the dataset to get the intermediates right before the post-adaption?
2. Can the proposed post-adaption cover all the reactions?

---

> ### Author Response · Authors · 2023-11-21
>
> Thank you very much for acknowledging our work! In response to your concerns, we would like to offer the following response.
>
> ---
>
> **Regarding the two identified weaknesses:**
> > W1: The diffusion models usually require more time. In this paper, the two-stage diffusion models could be less efficient than other baselines in terms of the inference time.
>
> Admittedly, diffusion models have a common drawback compared to other end-to-end models. However, the inefficiency of the two-stage diffusion model inference is not as pronounced as it may seem, thanks to task specificity. For external bond generation, we require only a few steps to complete the inference (as few as 50 steps in the paper's setting), owing to the small distributional difference between the two sides.
>
> > W2: For the top-1 accuracy, the performance gain of the proposed method is marginal. However, it outperforms in top-3, 5, and 10 accuracy.
>
> Although there is only a marginal improvement in top-1 accuracy, it is essential to recognize that the significance of top-3/5/10 accuracy is on par with that of top-1. Consequently, the overall competitiveness of our method remains robust.
>
> ---
>
> **Regarding the two questions:**
>
> > Q1: How to preprocess the dataset to get the intermediates right before the post-adaption?
>
> For external groups: The dataset employs a numbered mapping of atoms that remain unchanged before and after the reaction.
> Consequently, external groups can be automatically distinguished by identifying unnumbered atoms. For external bonds: After removing the external group from the product, a direct comparison between the product and the reactant allows us to identify the broken bond. All these operations can be carried out using RDKit.
>
> > Q2: Can the proposed post-adaption cover all the reactions?
>
> It can cover approximately 95% of reactions (we have verified that the percentage of special cases is within reasonable bounds).
>
> ---
> We hope that the provided information addresses your concerns.
>
> Best regards.

---

### Official Review · Reviewer_rjec · 2023-10-30

**Soundness:** 2 fair
**Presentation:** 1 poor
**Contribution:** 3 good
**Rating:** 3
**Confidence:** 3

**Summary:**

The paper presents a method for generating reactants of a product as two-stage diffusion model: first one generates additional atoms needed, and then connects them with the product.

**Strengths:**

- The main idea of predicting in two stages is good, and connects seemingly to how reactions work. The first stage can be seen as gathering the required extra atoms, and second stage is motivated by finding the reaction center. This part could have been strenghtened with more chemical analysis and motivation.
- The performance is competetive, but perhaps a bit inconclusive.

**Weaknesses:**

- The paper is poorly written, and is not up to par of ICLR publications. The math suffers from adhoc presentation wrt densities, distributions and mappings. I'm not convinced the method is mathematically correctly presented.
- The method is incremental: the diffusion models are taken off-the-shelf, and also the overall approach of group/bond generation is already seemingly known. It’s difficult to see why this method works well, and I suspect it’s mostly transformer tuning. This is not elaborated in the paper. Finally, the method is poorly motivated and novelties are vague. I think this is a succesful engineering effort towards solving an important problem, but scientifically there is very little contribution.

**Questions:**

- The introduction is vaguely written and I couldn’t follow what are the open problems this work tackles, or what are its contributions or their motivations. Many sentences are just impossible to understand. A good example of this is the incomprehensible sentence: “However, directly following the chemical reaction tem-plate could render the property of exploration on the distribution transformation of the diffusion model also be contrast to the data inherent structure by adding artifacted order to the motif and bonds.” (??) The intro also claims that autoregressive models are bad (yet this paper proposes a diffusion model), and that accumulating errors is somehow good. I don’t understand what the paper means by “reset”.
- The f is unlikely to be invertible, so the notation f^{-1} seems quite adhoc.
- The f(p()) notation seems nonsensical. You are mapping a density value to something. This does not seem to make sense. Similarly, the f( p*p/P_X ) is also nonsense: you can’t divide a density evaluation with a distribution, and then map it to something.
- I don’t understand fig2. So we start with 3 balls in the left: what are these? Then they become 3 colored balls with one edge. What does this mean? Are the balls atoms and colors the atom types?
- How do you know how many extra atoms to add (the external group)? What does “noisy” external group mean?
- Eq 6 feels nonsensical: what does it mean to have a loss between x and p(x)? One is a graph, another is a scalar.

---

> ### Author Response · Authors · 2023-11-21
> **Response to the weaknesses**
>
> Thanks so much for your comments and constructive suggestions! We will address your concerns in the following paragraphs. Hope these replies could resolve your concerns, and any further comments are welcome!
>
> ---
>
> > W1: The paper is poorly written, and math suffers from adhoc presentation wrt densities, distributions and mappings.
>
> Thank you for meticulously scrutinizing the mathematical details of the paper! We have made substantial revisions to the mathematical aspects, primarily in section 2, to guarantee the accuracy of formulations and facilitate ease of comprehension in formula derivations.
>
> ---
>
> > W2: The method is incremental: the diffusion models are taken off-the-shelf, and also the overall approach of group/bond generation is already seemingly known. It’s difficult to see why this method works well, and I suspect it’s mostly transformer tuning. This is not elaborated in the paper. Finally, the method is poorly motivated and novelties are vague. I think this is a successful engineering effort towards solving an important problem, but scientifically there is very little contribution.
>
> Thank you for highlighting the motivational and creative nuances in our approach. We have refined the introductory section to enhance the clarity and robustness of our motivation. In essence, our motivation unfolds in two key dimensions:
>
> 1. Enhancing Semi-Template Methods: Semi-template methods offer scalability[1,2] (compared to template-based approaches), interpretability and diversity[3] (compared to template-free methods). However, current semi-template methods fall short in performance. Our goal is to pioneer a suite of more efficient semi-template methods, capitalizing on their inherent advantages.
>
> 2. Innovating with Diffusion Models for Retrosynthesis: Conceptualizing the retrosynthesis as a distribution interpolation challenge, we identify diffusion models as a promising architecture for probabilistic modeling. However, existing templates overly constrain the intrinsic data structure, necessitating artificial modifications to molecular structures—a hindrance to feasible distribution transformations. To overcome this, we have redefined the templates, tailoring them to the modeling requisites of distribution transformations.
>
> Addressing this complex problem goes beyond a successful engineering effort, as employing diffusion models directly proves impractical. Transformer tuning alone lacks intrinsic effectiveness.
>
> Our scientific contribution lies in redefining the chemical template, aligning it with the modeling requirements of the task, thereby maximizing the capabilities of the diffusion model.
>
> ---
>
> **References**
>
> [1] Chaochao Yan, Qianggang Ding, Peilin Zhao, Shuangjia Zheng, Jinyu Yang, Yang Yu, and Junzhou
> Huang. Retroxpert: Decompose retrosynthesis prediction like a chemist. Advances in Neural
> Information Processing Systems, 33:11248–11258, 2020.
>
> [2] Chence Shi, Minkai Xu, Hongyu Guo, Ming Zhang, and Jian Tang. A graph to graphs framework
> for retrosynthesis prediction. In International conference on machine learning, pp. 8818–8827.
> PMLR, 2020.
>
> [3] Benson Chen, Tianxiao Shen, Tommi S Jaakkola, and Regina Barzilay. Learning to make general-
> izable and diverse predictions for retrosynthesis. arXiv preprint arXiv:1910.09688, 2019.

---

> ### Author Response · Authors · 2023-11-21
> **Response to the questions**
>
> > Q1: The introduction is vaguely written and I couldn’t follow what are the open problems this work tackles, or what are its contributions or their motivations. Many sentences are just impossible to understand. The intro also claims that autoregressive models are bad (yet this paper proposes a diffusion model), and that accumulating errors is somehow good. I don’t understand what the paper means by “reset”.
>
> Thank you for highlighting the crucial aspects concerning our contributions and motivations. We have meticulously revised the introductory section to amplify both clarity and robustness in articulating our motivation. In response to W2, we have provided detailed elaborations.
>
> To address incomprehensible sentences, including the one you flagged, we have taken proactive measures to rectify them. Furthermore, we acknowledge and rectify the inappropriate inclusion of the cumulative error of the autoregressive model as a motivation, opting for its removal.
>
> Additionally, the term "reset" has been accurately adjusted to "redefine" to signify our deliberate redefinition of the chemical prior. This adjustment underscores our commitment to tailoring the model to the distributional transitions, making a distinct contribution to the field of semi-template methods.
>
> > Q2: The f is unlikely to be invertible, so the notation f^{-1} seems quite ad hoc.
>
> Thanks for pointing out our laxity in mathematical formalization. We have removed the ambiguous mappings f and f^{-1} to make the task definition clearer (see Section 2 for modified details).
>
> > Q3: The f(p()) notation seems nonsensical. You are mapping a density value to something. This does not seem to make sense. Similarly, the f( p*p/P_X ) is also nonsense: you can’t divide a density evaluation with a distribution, and then map it to something.
>
> Thanks for pointing out our laxity in mathematical formalization. We have rewritten notations of the task definition and the template setup (see Section 2 for modified details), you can check it and feel free to ask new questions if any.
>
> > Q4: I don’t understand fig2. So we start with 3 balls in the left: what are these? Then they become 3 colored balls with one edge. What does this mean? Are the balls atoms and colors the atom types?
>
> Thanks for pointing out the lack of clarity in our presentation. We have modified the interpretation of Figure 2 to make the information in the figure more understandable. Specifically, balls are atoms and colors are atom types. The 3 white balls on the left mean 3 dummy atoms, and the colored balls mean the 3 dummy atoms have been denoised into real atom categories. About the elaboration of dummy atoms, you can see section 2.2 (have updated) for more details.
>
> > Q5: How do you know how many extra atoms to add (the external group)? What does “noisy” external group mean?
>
> Thanks for pointing out the lack of clarity in our presentation. The number of extra atoms is set as a constant, and we set a dummy atom category for indirectly judging the actual number (for example, the unexisted atom is set as the dummy category, you can see updated section 2.2 for details). The"noisy" external group are all dummy atoms with no bonds.
>
> > Q6: Eq 6 feels nonsensical: what does it mean to have a loss between x and p(x)? One is a graph, another is a scalar.
>
> Thanks for pointing out the presentation ambiguity. Eq 6 (now Eq 8) was intended to calculate the cross-entropy of the ground truth with the logic output by the network, but there is ambiguity in formalization. We have modified that loss function to avoid ambiguity, as shown in Eq 8.

---

> ### Author Response · Authors · 2023-11-23
>
> Dear Reviewer,
>
> We've already submitted our response, and only a few hours are left before the deadline. Does it address your questions? We are more than happy to answer any further questions.
>
> Thanks!

---

### Official Review · Reviewer_iBeY · 2023-10-31

**Soundness:** 2 fair
**Presentation:** 1 poor
**Contribution:** 2 fair
**Rating:** 3
**Confidence:** 4

**Summary:**

This paper proposes RetroDiff: a model casting retrosynthesis prediction as a two-stage discrete diffusion process. In the denoising direction, the first stage generates the missing atoms (those that were part of side products and hence are not seen on the product side) and then the second stage generates the bonds between the product and the added atoms. Finally, there is a final (unlearned) stage, which breaks at most one pre-existing bond. The authors experiment on USPTO-50K as well as analyse the behaviour of their method qualitatively.

**Strengths:**

(S1): The idea to apply discrete diffusion to reaction prediction is, to my knowledge, novel (or at least doesn't appear in any established work). It also seems potentially promising given recent advances in diffusion models overall.

(S2): The paper includes some useful qualitative visualizations (Figures 5-6), which help get intuition about how RetroDiff works.

**Weaknesses:**

(W1): The empirical results are not too impressive. In the paper, the results appear more promising than they actually are, as several baselines are missing, and the presentation is also somewhat biased.

- (a) Strong baselines are missing: for example RetroKNN [1] on the template-based side, and RootAligned SMILES [2] on the template-free side. Both of these models outperform RetroDiff by a large margin (a few %) across top-k values. Note that recent work [3] corrected some of the previously reported results on USPTO-50K, showing that e.g. results for LocalRetro and RetroKNN were somewhat inflated, but even after the correction RetroKNN is stronger than RetroDiff.

- (b) Only accuracy with top-k for k <= 10 is reported, while it is standard to also report top-50. Seeing unpromising scaling from top-5 to top-10, I imagine top-50 is not very impressive, but it should nonetheless be reported, to make sure the downsides of the presented method are explicitly shown. Although top-50 doesn't differentiate well between some models as they approach 100% on simple datasets such as USPTO-50K, it is still important, because it shows whether the model is able to at least in principle cover the full distribution. Also, popular multi-step search approaches often query the single-step model for up to 50 reactions.

- (c) The authors focus their comparison within semi-template methods, noting how their preformance is strong within that class. This to me seems a bit artificial, as belonging to the semi-template class doesn't convey strong benefits in itself that would warrant not looking at other model classes. One could argue that template-free methods are too unconstrained and not interpretable, and hence not focus on them in a potential comparison, but template-based methods are usually at least as interpretable as semi-template-based ones, and sometimes more intepretable (because interpretablity is usually implemented by looking up the literature reactions that gave rise to a particular template that got applied, and this method is not applicable to some semi-template-based methods like RetroDiff). In consequence, I mostly looked at performance among all reaction models overall, and in that setting RetroDiff doesn't perform too favourably.

- (d) Finally, the authors also show improved SMILES validity score compared to popular models based on the Transformer architecture. This is a good sanity check, although I wouldn't put too much weight into that result, as many highly performant template-based models (e.g. LocalRetro, RetroKNN) have 100% validity by design. It is still good to see RetroDiff getting near-perfect validity, but it's important to remember that this limitation is actually absent in some model classes.

(W2): In Section 2.1.3 on post-adaptation, the authors explain that in the final stage, zero or one bond is broken in the transformed product graph to form the reactant graphs. To me, this seems to assume there are at most two reactants. This may be true in USPTO-50K, but would not be true overall. While the number of reactants is usually small, in larger reaction databases one can find a lot of examples with 3 or even 4 reactants. Am I correct in assuming that the current post-adaptation procedure only works if there are 2 reactants? If so, this limitation should be highlighted, as it would be a significant hindrance to real-world usability.

(W3): Several parts of the work are not clear, which prevent full understanding of some of the modelling or algorithmic details – see the "Questions" section for concrete examples. Finally, the writing needs improvement to make the paper reasonably easy to read (see the "Nitpicks" section below for concrete suggestions).

=== Other comments ===

(O1): The paper (e.g. in "Related Works" section) mentions "retrosynthesis planning", but this is usually understood as planning multi-step chemical syntheses, which is not what the paper is about. It may be more precise to say "retrosynthesis prediction" or "backward reaction prediction".



=== Nitpicks ===

Below I list nitpicks (e.g. typos, grammar errors), which did not have a significant impact on my review score, but it would be good to fix those to improve the paper further.

- "pathways through a given product" -> "to"?

- "chemical priori" (in two places) -> "chemical prior"

- "limited chemical reaction diversity and interpretability render the potential" -> perhaps you meant that these issues "hinder" the potential, not "render"; the word "render" is misused like this in a few places

- Many words are unnecessarily capitalized e.g. those appearing after a semicolon, inside parenthesis or after a comma

- Sentence starting in "However, directly following the chemical reaction (…)" is very long and hard to parse, I would suggest revising it, also fixing phrases like "also be contrast". Finally, "artifacted" should instead be "artificial".

- "contributions are the following three-fold" -> either "the following" or "three-fold", not both

- "connect the given product and the justly generated group" -> I would replace "justly" with "just" or "recently"

- "we aims to" -> "aim"

- "with a training step of 100000" -> I guess you mean _number_ of training steps?

- "Despite the efficiencies of data-driven" -> missing "methods" at the end?

- The word "reset" is misused in many places, often used to mean "redefine"



=== References ===

[1] Xie at el, "Retrosynthesis Prediction with Local Template Retrieval"

[2] Zhong et al, "Root-aligned SMILES: A Tight Representation for Chemical Reaction Prediction"

[3] Maziarz et al, "Re-evaluating Retrosynthesis Algorithms with Syntheseus"

**Questions:**

(Q1): Is Equation 4 fully mathematically correct? I am a bit confused by the notation and the disappearance of the integral.

(Q2): Could you elaborate on "We have obtained a trained network $p_θ$ in the last stage, so we freeze g and x in the graph and continue to train $p_θ$."? Is the overall networks trained in stages utilizing different training "datasets"?

(Q3): Could you elaborate on Section 2.2? I feel like the reader needs to fill in a lot of details to understand it, e.g. infer the exact structure of $a_1$, $b_1$ and $b_2$ (how many ones/zeros); it is also confusing that these are used to define $v_1$ and $v_2$ before being properly introduced themselves.

---

> ### Author Response · Authors · 2023-11-21
> **Response to the presentation (W3, nipticks, Q1-Q3 in the original comment)**
>
> Thank you for your detailed writing suggestions. We have carefully addressed the points in your nitpick list, making revisions to enhance the overall quality of our writing. Regarding your three questions, we provide the following explanations:
>
> > Q1: Is Equation 4 fully mathematically correct? I am a bit confused by the notation and the disappearance of the integral.
>
> Thank you for meticulously scrutinizing the mathematical details of the paper! We have rewritten the "Denoising" part in section 2.1.1. The mathematical derivation is now more detailed and accurate.
>
> > Q2: Could you elaborate on "We have obtained a trained network $p_{\theta}$ in the last stage, so we freeze g and x in the graph and continue to train $p_{\theta}$."? Are the overall networks trained in stages utilizing different training "datasets"?
>
> Thanks for pointing out the lack of clarity in our presentation. To be precise, different targets were employed, as elaborated below:
>
> In the first stage, the input comprises a graph with a noisy external group and the true product, where no bond exists between the two (i.e., a dummy category). The output is the true external group.
>
> In the second stage, the input is also a graph, but now it consists of the true external group and the true product, with noisy external bonds connecting the two. The output is the true external bond.
>
> It's important to note that the graph structure remains identical in the two stages, with the only difference being whether different parts of the input graph are noisy or true.
>
> > Q3: Could you elaborate on Section 2.2? I feel like the reader needs to fill in a lot of details to understand it, e.g. infer the exact structure of a1, b1 and b2 (how many ones/zeros); it is also confusing that these are used to define v1 and v2 before being properly introduced themselves.
>
> Thanks for pointing out the lack of clarity in our presentation. We have rewritten section 2.2 to make it easy to understand. Ambiguous notations, such as a1, b1, and b2, have been replaced, and detailed explanations have been added to enhance clarity.

---

> ### Author Response · Authors · 2023-11-21
> **Response to the experimental comparisons (W1 in the original comment)**
>
> > (a) Strong baselines are missing, and RetroKNN (template-based), RootAligned (template-free) is stronger than RetroDiff, and they are stronger than RetroDiff.
>
> Thanks a lot for mentioning the related works. It should be noted that the scope of our work focuses on the semi-template methods where tradeoff between the scalability and interpretability is emphasized. As the reviewer suggested, we have added the mentioned baselines and the discussion on their difference in the model class with our model in the updated version for the completeness of the empirical comparison.
>
> As you noticed, in terms of performance, our method falls short of the template-based RetroKNN. However, our approach attains the SOTA performances in the semi-template methods and holds a noteworthy advantage over the template-based methods. For more details, please refer to the response to (c).
>
> ---
> > (b) Only accuracy with top-k for k <= 10 is reported, while it is standard to also report top-50, to make sure the downsides of the presented method are explicitly shown.
>
> Thank you for the suggestion of involving more evaluation metrics and we agree that the distributional coverage performance of the method should also be explicitly discussed.  We also include the top-50 accuracy evaluation with a comparison to other methods(mainly template-based, as most semi-template method does not include the metric) in the following Table. We could find that the proposed method still shows competitive performance in the distributional coverage even compared to advanced templated-based methods.
>
> Template-based:
>
> | Retrosim | neuralsym | GLN | LocalRetro | RetroKNN |
> | --- | --- | --- | --- | --- |
> | 85.3 | 83.1 | 92.4 | 97.7 | 96.5 |
>
> Semi-template:
> | RetroXpert | MEGAN | RetroDiff |
> | --- | --- | --- |
> | 64.0 | 93.2 | 88.6 |
>
> ---
> > (c) S
> emi-template class doesn't convey strong benefits. Template-based methods are usually at least as interpretable as semi-template-based ones, and sometimes more interpretable. The performance should compare all reaction models overall but not only semi-template methods, and in that setting RetroDiff doesn't perform too favourably.
>
> - Firstly, we acknowledge that the performance of Retrodiff is not on par with the current SOTA template-based method, Retroknn. However, the difference is not insignificant (retroknn's top-1 result is 55.3%, 2.7% higher than ours). Furthermore, it is crucial to reiterate that we lead as the SOTA among semi-template methods.
>
> - Nevertheless, the most significant drawback of the template-based approach, as we have previously highlighted in the introduction, is its inherent scalability disadvantage. This limitation has been extensively discussed in prior research, and we substantiate this point by citing arguments from other papers:
>
>   - RetroXpert[1]: "The reported synthetic organic knowledge consists of in the order of 10^7 reactions and compounds[5]... It is infeasible to manually encode all the synthesis routes in practice considering the exponential growth in the number of reactions[6]... An obvious limitation is that these methods can only infer reactions within the chemical space covered by the template database, preventing them from discovering novel reactions[7]."
>
>   - G2G[2]: "Despite their great potential for synthesis planning, template-based methods, however, not only require expensive computation but also suffer from poor generalization on new target structures and reaction types."
>
>   - Retroformer[3]: "Despite the state-of-the-art accuracy and guaranteed molecule validity, these methods are limited to the scope of the existing template database."
>
>   - MEGAN[4]: "Due to computational limitations, they typically require representing reactions using a restricted number of templates, which necessarily limits the coverage of the chemical reaction space accessible by such methods."
>
> - In general, stepping back to survey the landscape of prior research, the template-based approach indeed boasts performance advantages, but its notable scalability issue remains a persistent challenge. Conversely, the semi-template approach stands out with two key strengths: scalability and interpretability, making a distinct and valuable contribution. These two approaches can run in parallel rather than as opposing forces, each bringing significant value to the research community. We hope our contributions are duly acknowledged. Thank you once again!
> ---
> > (d) Although RetroDiff getting near-perfect validity, the sanity check is not so important, because many highly performant template-based models (e.g. LocalRetro, RetroKNN) have 100% validity by design.
>
> Thank you for bringing this to my attention! I acknowledge that the template-based approach excels in achieving legitimacy due to the comprehensive templates. Nevertheless, our RetroDiff approach is also highly proficient in this aspect. While not a point for celebration, it underscores the robustness of our model.

---

> > ### Author Response · Authors · 2023-11-21
> > **References**
> >
> > [1] Chaochao Yan, Qianggang Ding, Peilin Zhao, Shuangjia Zheng, Jinyu Yang, Yang Yu, and Junzhou
> > Huang. Retroxpert: Decompose retrosynthesis prediction like a chemist. Advances in Neural
> > Information Processing Systems, 33:11248–11258, 2020.
> >
> > [2] Chence Shi, Minkai Xu, Hongyu Guo, Ming Zhang, and Jian Tang. A graph to graphs framework
> > for retrosynthesis prediction. In International conference on machine learning, pp. 8818–8827.
> > PMLR, 2020.
> >
> > [3] Yue Wan, Chang-Yu Hsieh, Ben Liao, and Shengyu Zhang. Retroformer: Pushing the limits of
> > end-to-end retrosynthesis transformer. In International Conference on Machine Learning, pp.
> > 22475–22490. PMLR, 2022.
> >
> > [4] Mikołaj Sacha, Mikołaj Błaz, Piotr Byrski, Paweł Dabrowski-Tumanski, Mikołaj Chrominski, Rafał
> > Loska, Paweł Włodarczyk-Pruszynski, and Stanisław Jastrzebski. Molecule edit graph attention
> > network: modeling chemical reactions as sequences of graph edits. Journal of Chemical Infor-
> > mation and Modeling, 61(7):3273–3284, 2021.
> >
> > [5] Chris M Gothard, Siowling Soh, Nosheen A Gothard, Bartlomiej Kowalczyk, Yanhu Wei, Bilge Baytekin, and Bartosz A Grzybowski. Rewiring chemistry: Algorithmic discovery and experimental validation of one-pot reactions in the network of organic chemistry. Angewandte Chemie International Edition, 51(32):7922–7927, 2012.
> >
> > [6] Marwin HS Segler, Mike Preuss, and Mark P Waller. Planning chemical syntheses with deep neural networks and symbolic ai. Nature, 555(7698):604–610, 2018.
> >
> > [7] Marwin HS Segler and Mark P Waller. Modelling chemical reasoning to predict and invent reactions. Chemistry–A European Journal, 23(25):6118–6128, 2017.

---

> ### Author Response · Authors · 2023-11-21
> **Response to others (W2 in the original comment)**
>
> **Regarding the generalizability of post-adaptation to more complex retrosynthesis datasets:**
>
> The adaptability of post-adaptation is scalable. To illustrate, when dealing with 3 reactants, there are 4 reaction sites. Therefore, it becomes crucial to determine whether these 4 sites form two sets of adjacent sites to establish their legitimacy. This principle holds true for scenarios involving more than 3 reactions.

---

> ### Author Response · Authors · 2023-11-23
>
> Dear Reviewer,
>
> We've already submitted our response, and only a few hours are left before the deadline. Does it address your questions? We are more than happy to answer any further questions.
>
> Thanks!

---

> > ### Comment · Reviewer_iBeY · 2023-11-23
> > **Response to rebuttal**
> >
> > Thanks for the detailed rebuttal. To summarize my final thoughts:
> > - *Clarity:* I think the paper has been improved in this regard.
> > - *Results only being good in the semi-template model realm:* I see your point about potential worse scalability of template-based models, yet this only matters if a template-based model starts performing badly at a large scale, and is overtaken by another model type. Thus, I wouldn't necessarily assume RetroDiff would work better on a large scale dataset than a template-based model without experiments to back this up.
> > - *Post-adaptation scaling to more complex reactions:* I see how you could potentially generalize your approach, but are you saying the result would always be deterministic, i.e. the post-adaptation would result in a single outcome rather than several? My feeling is that, with increasing complexity, there will be more possibilities to "fix up" the pre-existing bonds (I can be wrong though). At the very least, it is not clear to me how precisely the post-adaptation algorithm would look like if an arbitrary number of bonds were broken.

---

### Official Review · Reviewer_1Pis · 2023-11-22

**Soundness:** 3 good
**Presentation:** 2 fair
**Contribution:** 3 good
**Rating:** 5
**Confidence:** 5

**Summary:**

a new diffusion model for retrosynthesis is presented

**Strengths:**

- diffusion has not been applied much to retrosynthesis
- results are decent but not impressive

**Weaknesses:**

- several baselines were missing
- the design of the model seems to be very complex
- inference times are not reported
-citation of graph2edit is missing, which is semi-template, and very strong https://www.nature.com/articles/s41467-023-38851-5

**Questions:**

- why are so many complex steps and rule checking needed? a good generative model would learn all of that from the data. have the author ablated this properly?
- in the current form, the paper is a bit unmotivated. will the community just apply any new generative model to retrosynthesis? autoregressive -> GANs -> VAE -> diffusion -> next is flow matching I guess?

**Details Of Ethics Concerns:**

-

---

> ### Author Response · Authors · 2023-11-23
>
> Thanks so much for your constructive comments! We will address your concerns from three perspectives according to your comments.
>
> ---
>
> **Regarding model design complexity:**
>
> We didn't design very complicated models. Instead, we defined a new chemical template to implement the simplest distributional transformations on this basis.
>
> The previous templates imposed numerous constraints, demanding human intervention in atom and bond manipulation during reactions, which causes the incompatibility of generative models. We redefined the templates, breaking down the task into two serial stages: external group generation and external bond generation. This two-stage approach simplifies the problem and aligns it with the application scenario of diffusion modeling, allowing for the direct learning of distributions.
>
> And the last step of RULE CHECKING is just a simple check based on the legitimacy of the chemical compounds, and is not a complex predefined rule.
>
> ---
>
> **Regarding motivation:**
>
> We have clearly elaborated the motivation in our response to **reviewer rjec**:
>
> 1. Enhancing Semi-Template Methods: Semi-template methods offer scalability[1,2] (compared to template-based approaches), interpretability and diversity[3] (compared to template-free methods). However, current semi-template methods fall short in performance. Our goal is to pioneer a suite of more efficient semi-template methods, capitalizing on their inherent advantages.
>
> 2. Innovating with Diffusion Models for Retrosynthesis: Conceptualizing the retrosynthesis as a distribution interpolation challenge, we identify diffusion models as a promising architecture for probabilistic modeling. However, existing templates overly constrain the intrinsic data structure, necessitating artificial modifications to molecular structures—a hindrance to feasible distribution transformations. To overcome this, we have redefined the templates, tailoring them to the modeling requisites of distribution transformations.
>
> Addressing this complex problem goes beyond a successful engineering effort, as employing diffusion models directly proves impractical. Transformer tuning alone lacks intrinsic effectiveness. Our scientific contribution lies in redefining the chemical template, aligning it with the modeling requirements of the task, thereby maximizing the capabilities of the diffusion model.
>
> ---
>
> **Regarding baseline:**
>
> Thanks for pointing out the lack of a baseline. We apologize for neglecting this recent work, and we will discuss it further in the related work.
>
> ---
>
> Hope these replies could resolve your concerns, and any further comments are welcome!
>
> **References**
>
> [1] Chaochao Yan, Qianggang Ding, Peilin Zhao, Shuangjia Zheng, Jinyu Yang, Yang Yu, and Junzhou
> Huang. Retroxpert: Decompose retrosynthesis prediction like a chemist. Advances in Neural
> Information Processing Systems, 33:11248–11258, 2020.
>
> [2] Chence Shi, Minkai Xu, Hongyu Guo, Ming Zhang, and Jian Tang. A graph to graphs framework
> for retrosynthesis prediction. In International conference on machine learning, pp. 8818–8827.
> PMLR, 2020.
>
> [3] Benson Chen, Tianxiao Shen, Tommi S Jaakkola, and Regina Barzilay. Learning to make general-
> izable and diverse predictions for retrosynthesis. arXiv preprint arXiv:1910.09688, 2019.

---

### Author Response · Authors · 2023-11-22
**General Response**

We thank all reviewers for their valuable comments! Overall, the reviewers' main concerns were two-fold:

**(1) Writing Details**

The initial paper draft contains numerous unclear elaborations and imprecise mathematical derivations, mainly in the introduction and method sections of the paper. We have diligently addressed these concerns by extensively revising both sections, incorporating corrections, and addressing all the feedback provided by the reviewers.

**(2) Experimental Comparisons**

**Reviewer iBeY** pointed out our incomplete experimental comparison. He argued that the semi-template method does not have a unique advantage and therefore needs to be fully compared with all model classes. This perspective is actually a long-standing open discussion in this field about the strengths and weaknesses of various model classes. In fact, semi-template methods have the advantages of scalability (compared to template-based) versus interpretability and diversity (compared to template-free), and have been showing good growth, even though their current SOTA performance is not as good as that of template-based methods. We think this is a point that deserves further discussion, and not to be Achilles' heel.

We hope that our response and revised paper will address all the issues!

---

### Meta-Review · Area_Chair_LMDj · 2023-12-05

**Metareview:**

The reviewers found this work topical and interesting. However, all reviewers had concerns related to the complicated setup, empirical results not being convincing, and issues with clarity and presentation. Even if the authors did a good job during the rebuttal phase in clarifying things, the reviewer consensus reads that this paper requires more work on all these aspects, and it should go through a 'major revision'. Thus, the decision is to reject this work in its current form.

**Justification For Why Not Higher Score:**

This paper could be interesting for the audience of ICLR and is certainly very topical. There are several papers submitted to ICLR on diffusion for reaction modelling, and all of them seem to propose interesting ideas for the problem. Even this one. However, I agree with the reviewers that this paper has several issues.

**Justification For Why Not Lower Score:**

N/A

---

### Decision · Program_Chairs · 2024-01-16

Reject